# Semi-supervised Multimodal Coreference Resolution in Image Narrations

**Arushi Goel[1], Basura Fernando[2], Frank Keller[1], and Hakan Bilen[1]**
[1]School of Informatics, University of Edinburgh, UK
[2]CFAR, IHPC, A*STAR, Singapore

## Abstract

In this paper, we study multimodal coreference resolution, specifically where a longer descriptive text, *i.e.*, a *narration* is paired with an image. This poses significant challenges due to fine-grained image-text alignment, inherent ambiguity present in narrative language, and unavailability of large annotated training sets. To tackle these challenges, we present a data efficient semi-supervised approach that utilizes image-narration pairs to resolve coreferences and narrative grounding in a multimodal context. Our approach incorporates losses for both labeled and unlabeled data within a cross-modal framework. Our evaluation shows that the proposed approach outperforms strong baselines both quantitatively and qualitatively, for the tasks of coreference resolution and narrative grounding.

## 1 Introduction

In linguistic processing, coreference resolution is a standard task that aims to identify referring expressions such as noun phrases and pronouns that refer to the same entity. It is fundamental to many standard problems including question answering (Kwiatkowski et al., 2019; Das et al., 2017), sentiment analysis (Cambria et al., 2017; Medhat et al., 2014), summarization (Gupta and Lehal, 2010; Shi et al., 2021) and machine translation (Lopez, 2008; Bahdanau et al., 2014; Wu et al., 2016). In this work, we focus on a multimodal coreference resolution (MCR) scenario where the coreferences occur in a narration paired with an image and also link to an image region as shown in Figure 1. Here resolving coreferences is challenging, as mentions referring to different entities can be very similar when encoded by a language model, *e.g.*, *one boy*, *the other boy*, *the boy*. Hence it demands a fine-grained understanding of each modality and as well as across them. In particular, it requires simultaneously grounding instances by

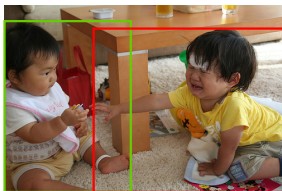

we can see **two small boys** are sitting on a white color mat and in that **one boy** is crying and **he** is wearing a yellow color t-shirt and grey color short. **The other boy** is wearing white color t-shirt and cream color short and **he** is also holding some object in **his** hand. On the head of **the boy** we can see a white color sticker and on **their** t-shirts we can see some text and designs also.

Figure 1: Example image-narration pair from the Coreferenced Image Narratives dataset (Goel et al., 2022). Phrases marked in the same color corefer to the same entity which are also grounded in the image. We do not show singletons for brevity.

identifying fine-grained visual details (*e.g.*, disambiguating them by recognizing the action 'crying', spotting 'white color t-shirt and cream color short' or 'a white color sticker on the head'), and capturing long-range dependency across sentences (*e.g.*, *two small boys* and *their*).

MCR has recently gained increasing attention, with several notable studies (Ramanathan et al., 2014; Huang et al., 2018; Cui et al., 2021; Parcalabescu et al., 2021; Das et al., 2017; Guo et al., 2022; Goel et al., 2022; Hong et al., 2023). However, many of them focus on images with simple short sentences, such as 'A woman is driving a motorcycle. Is she wearing a helmet?' (Das et al., 2017; Parcalabescu et al., 2021), or are limited to identifying movie characters or people (Ramanathan et al., 2014; Cui et al., 2021). More recently, Goel et al. (2022) introduced a challenging and unconstrained MCR problem (see Figure 1) including a dataset, Coreferenced Image Narratives (CIN), with both people and objects as referents with long textual descriptions (narrations). As manually annotating a large dataset with coreferencing and grounding labels is expensive, the authors provide annotations only for evaluation purposes. They also propose a weakly supervised method that learns to jointly ground mentions in images and use them as anchors along with prior linguistic rules (Lee et al., 2011) to group coreferring men-

tions from only image and narration pairs without the annotations. The method has multiple shortcomings: (1) weakly supervised grounding fails to disambiguate multiple instances of the same object class, boy (*one boy*, *the other boy*), (2) language rules such as *exact match of phrases* are either too strict or too generic, *e.g.*, *pronoun match*, linking pronouns to one antecedent (*one boy*, *he*, *he*, *his*) and, (3) they require an additional modality, mouse traces to learn coreferences which can be expensive to obtain.

Motivated by these limitations, we argue that it difficult to successfully resolve coreferences from only image-narration pairs in cases where multiple instances of the same object category are present; this situation is common coincides with a mention in the narration. Since full manual annotations of coreference chains and bounding boxes is expensive, we propose to resolve coreferences and ground mentions in a semi-supervised setting where only a few data samples are labeled. Our approach involves a customized multi-modal fusion model that combines image region features and mention features from narrations through cross-attention (Vaswani et al., 2017; Li et al., 2021). We investigate different task-specific losses for training on labeled and unlabeled data, and show that naively combining training on the labeled and pseudo-labeled data suffers from severe overfitting (Arazo et al., 2020). Hence, we propose a robust loss function and thresholding-based training scheme to effectively learn from the unlabeled set. This novel approach results in consistent performance improvements with the inclusion of unlabeled data during training.

Our main contributions are (1) a vision-language framework for MCR trained on a small labeled and an unlabeled dataset, (2) novel task-specific losses (on both labeled and pseudo-labeled data) for learning joint multi-modal embeddings for coreference resolution while simultaneously improving narrative grounding, (3) extensive evaluation of our proposed method on the CIN dataset and ablation studies to validate our design choices, showing consistent performance gains compared to baselines on coreference resolution and narrative grounding.

## 2 Related work

**Multimodal coreference resolution.** MCR involves comprehending the contextual information in language and establishing connections with specific regions in an image. Recently, considerable efforts have been dedicated to developing datasets that can effectively address this intricate task. Parcalabescu et al. (2021) introduced the VALSE dataset, which encompasses various coreference scenarios. However, this dataset focuses on the downstream task of visual question answering without evaluating coreference resolution or grounding. Hence, we evaluate our method on CIN dataset (Goel et al., 2022) that contains coreference chains and grounding annotations. Another approach to MCR datasets involves linking people's names mentioned in the text to corresponding images and resolving pronouns that connect to those specific names (Ramanathan et al., 2014; Cui et al., 2021; Hong et al., 2023). However, our main focus is to resolve coreferences in a generic scenario (with visual complexity) unlike the others that are either limited to only people names/characters (Ramanathan et al., 2014; Cui et al., 2021; Hong et al., 2023) or have simple sentences (Das et al., 2017; Parcalabescu et al., 2021).

**Vision-language learning.** Existing work on vision and language understanding employs either pre-trained object detector features (He et al., 2017; Ren et al., 2015) as an image encoder, ViT (Dosovitskiy et al., 2020) or a CNN (Simonyan and Zisserman, 2014) combined with a transformer-based text encoder (Devlin et al., 2018). To model cross-modal interaction between the image and text encoders, UNITER (Chen et al., 2020), ALBEF (Li et al., 2021) and VinVL (Zhang et al., 2021b) employ a multimodal encoder. They are pre-trained on large-scale image-caption pairs such as COCO (Lin et al., 2014), Conceptual captions (Sharma et al., 2018; Changpinyo et al., 2021), Visual Genome (Krishna et al., 2017). The pre-training objectives are implemented with image-text contrastive loss, masked language modeling, and image-text matching loss. Our method is inspired by these architectures and is trained using a set of self-supervised and task-based objectives in a semi-supervised learning fashion.

**Semi-supervised learning.** There is a large body of work in semi-supervised learning (Zhai et al., 2019; Van Engelen and Hoos, 2020; Ouali et al., 2020). These methods typically exploit unlabeled data via either pseudo-labeling with small amounts of labeled data (Lee et al., 2013; Arazo et al., 2020; Rizve et al., 2021; Sohn et al., 2020; Zhang et al., 2021a) or by enforcing consistency regularization

(Berthelot et al., 2019; Abduweili et al., 2021) on the unlabeled data to produce consistent predictions over various perturbations of the same input by applying several augmentation strategies (Zhang et al., 2017; Cubuk et al., 2018, 2020). Our method draws inspiration from pseudo-labeling literature and uses a robust loss function and thresholding to counter overfitting to pseudo-labels.

## 3 Method

### 3.1 Task Overview

Our goal is (1) to group mentions (*i.e.*, referential words or phrases) in the narration that corefer to the same entity and, (2) to ground each mention to a region in an image. Formally, let $N = \{m_1, m_2, \ldots, m_{|N|}\}$ denote a narration with $|N|$ mentions for an image $I$ with $|I|$ regions where $I = \{r_1, r_2, \ldots, r_{|I|}\}$. We wish to learn an embedding function $f$ that takes in an image $I$ and its narration $N$, parsed to contain a set of mentions, and outputs a score for a mention pair $(m, m')$:

$$\frac{f(m) \cdot f(m')}{|f(m)||f(m')|} \quad (1)$$

The mention pair $m$ and $m'$ corefers if the score in Equation (1) is high, otherwise they do not.

For grounding of the mention $m$ on the image region $r$, we also learn another function $g$ that outputs a score for the mention $m$ being located at region $r$ in image $I$. Next, we describe in detail our methodology to learn the two functions $f$ and $g$.

### 3.2 Model Architecture

In Figure 2, we illustrate our model architecture. Each image is parsed into a set of regions through a pre-trained object detector (Ren et al., 2015), where each region $r$ is represented by a $d$-dimensional joint embedding $\boldsymbol{v}_r \in \mathbb{R}^d$ including its visual, semantic and spatial features. In particular, the visual encoder $f_v$ is instantiated as a transformer block that takes in a joint feature embedding $\boldsymbol{v}_r$ for the object region $r$ and outputs a $D$ dimensional embedding, *i.e.*, $f_v(\boldsymbol{v}_r) : \mathbb{R}^d \to \mathbb{R}^D$.

Furthermore, we encode the words in each narration $N$ using a tokenizer (Devlin et al., 2018) to get a set of tokens for the words $w \in \mathbb{R}^V$ where $V$ is the vocabulary size. The text encoder $f_t$ which is also a transformer block that takes in the word token $w$ and outputs a $D$ dimensional embedding, *i.e.*, $f_t(w) : \mathbb{R}^V \to \mathbb{R}^D$. The mention embeddings are computed by averaging its corresponding word

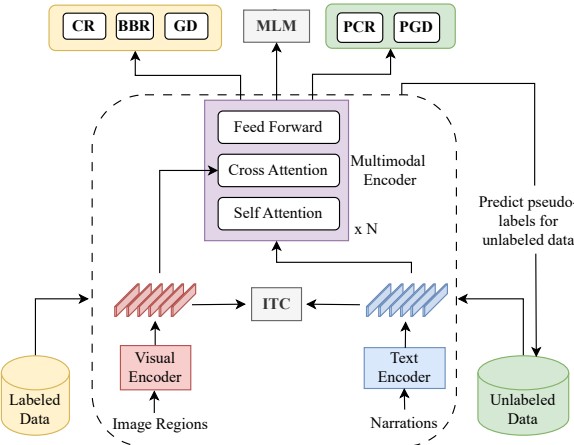

Figure 2: Illustration of our model architecture and training methodology. The pre-extracted image regions are fed into the visual encoder, the narrations are fed into the text encoder and both modalities are fused using a multimodal encoder. The model is optimized using self-supervised objectives (in grey) and specialized task-based losses on both the labeled data (in yellow boxes) and the pseudo-labeled data (in green boxes).

representations as: $f_t(m) = \frac{1}{|m|} \sum_{w \in m} f_t(w)$ where, $|m|$ indicates the mention length in words, and the embeddings $f_t(m)$ have the same dimensionality as the visual features.

Next, the multi-modal encoder $f$ fuses the visual features from the visual encoder $f_v(\boldsymbol{v}_r)$ with the mention features from the text encoder $f_t(m)$. Similar to the cross-modal architectures (Li et al., 2021; Zhang et al., 2021b), the embeddings from the text encoder are first encoded using self-attention layers (Vaswani et al., 2017). Then, a multi-head cross attention module integrates the textual and visual features. In the cross-attention module, the self-attended mention embeddings $f_t(m)$ are treated as the query, while the image representations $f_v(\boldsymbol{v}_r)$ are treated as keys and values. The attention weights between the mention $m$ and the region $r$ are given as:

$$g(m, r) = \frac{\exp\left(\frac{f_t(m)^T . f_v(\boldsymbol{v}_r)}{\sqrt{d}}\right)}{\sum_{r' \in I} \exp\left(\frac{f_t(m)^T . f_v(\boldsymbol{v}_{r'})}{\sqrt{d}}\right)} \quad (2)$$

where the softmax is computed over the image regions for each mention. This attention matrix (or the grounding function) $g$ from the multi-head cross attention learns fine-grained mention to region alignment scores. Finally, the vision-aware mention embedding is represented as:

$$f(m) = g(m, r) . f_v(\boldsymbol{v}_r) \quad (3)$$

where, $f(m) \in \mathbb{R}^D$. This weighted embedding is then passed to a feed-forward module (Li et al., 2021) with an MLP and layer normalization. All the transformer encoders/blocks are based on the architecture proposed by (Li et al., 2021). It is important to note that compared to Goel et al. (2022), our model fuses vision and text features with a multimodal encoder, unlike theirs.

### 3.3 Semi-supervised Learning

Concretely, we aim to learn the parameters of the modules $f_v$, $f_t$ and $f$ given a training dataset $\mathcal{D}$ with $|\mathcal{D}|$ samples of image-narration pairs. Specifically, we use a small labeled set $\mathcal{D}_s = \{x_i, y_i\}_{i=1}^{|\mathcal{D}_s|}$ where $x_i = \{I, N\}$ is the image-narration input pair and $y_i = \forall_{m \in N}\{P(m), A(m), b_m\}$ is the label for the input pair. In particular, the label for each mention $m$ in the narration is given as: $P(m)$ and $A(m)$, the set of positive and negative mentions respectively for the mention $m$ and $b_m$, the bounding-box coordinates of the region corresponding to the mention $m$.

Due to the unavailability of a large labeled training set, we leverage the unlabeled data $\mathcal{D}_u = \mathcal{D} \setminus \mathcal{D}_s$ where, $\mathcal{D}_u = \{x_i\}_{i=1}^{|\mathcal{D}_u|}$ with only image-narration pairs as inputs. Our overall training objective is the joint loss function as follows:

$$\sum_{(x,y) \in \mathcal{D}_s} \frac{1}{|\mathcal{D}_s|} \mathcal{L}_s(x, y) + \sum_{x \in \mathcal{D}_u} \frac{1}{|\mathcal{D}_u|} \mathcal{L}_u(x) \quad (4)$$

where, $\mathcal{L}_s$ is the supervised loss and $\mathcal{L}_u$ is the unsupervised loss. First, we discuss how to formulate task-based supervised losses on the dataset $\mathcal{D}_s$.

**(S1) Coreference loss (CR)** Specifically, we propose to learn the similarity between the mention embeddings using a supervised contrastive loss (Khosla et al., 2020) which is defined as:

$$\mathcal{L}_{cr} = \sum_{m \in N} \frac{-1}{|P(m)|} \sum_{p \in P(m)}$$
$$\log \frac{exp(f(m).f(p)/\tau)}{\sum_{a \in A(m)} exp(f(m).f(a)/\tau)} \quad (5)$$

where $\tau$ is the temperature. This loss helps to cluster embeddings for coreferring mentions together and push the embeddings of non-referrants away from each other.

**(S2) Grounding loss (GD)** To align the mention $m$ and region $r$, we use the grounding function $g$ defined in Equation (2). In particular, we first define the ground-truth binary alignment on the labeled

training set $\mathcal{D}_s$. For the ground-truth bounding box $b_m$ for a mention $m$ we compute the intersection over union (IoU) between this bounding-box and the $R$ pre-extracted image regions. This is crucial because we don't have the exact region-mention match for the detections from the object detector. Following this, we get the binary alignment function $h(m, r)$, which is 1 for the mention $m$ and the detected image region $r$ if the region $r$ has the maximum IoU overlap with the ground-truth bounding box $b_m$, and 0 otherwise. Once we have the ground-truth alignment $h(m, r)$, we compute the cross-entropy loss as:

$$\mathcal{L}_{gd} = -\sum_{m \in N} \sum_{r \in I} h(m, r) \log(g(m, r)) \quad (6)$$

**(S3) Bounding box regression loss (BBR)** We further propose to add additional supervision to refine the object proposals from the detector for a mention. For each mention $m$, the ground-truth bounding box localization is represented as $b_m = (x, y, w, h)$. To learn refinements, we predict the box deltas from the model as $\delta_m = (\delta_x, \delta_y, \delta_w, \delta_h)$ for each mention $m$. We then take the highest scoring region for a given mention $m$ as:

$$r_m = \arg\max_{r \in I} g(m, r). \quad (7)$$

Our goal is to learn a transformation that maps a proposed box $r_m$ to a ground-truth box $b_m$. We then apply the smooth-L1 loss following Ren et al. (2015), denoted as $\mathcal{L}_{bbr}$. Further details about this loss are given in the appendix.

Next, we discuss how to train on the unlabeled subset of the dataset by generating pseudo-labels for the coreference and grounding tasks.

**(U1) Pseudo coreference loss (PCR)** Given the unlabeled dataset $\mathcal{D}_u$, we compute the pseudo coreferring pairs for the mentions in $N$. More specifically, we compute pseudo-positives $\hat{P}(m)$ and pseudo-negatives $\hat{A}(m)$ for a mention $m$ by computing the cosine similarity between the embeddings as in Equation (1). For each mention $m$, if the similarity with another mention $m'$ is greater than a threshold then we label it as a positive otherwise a negative. Finally, we compute the triplet loss as:

$$\mathcal{L}_{pcr} =$$
$$\sum_{m \in N} \max(||f(m) - \frac{1}{|\hat{P}(m)|} \sum_{p \in \hat{P}(m)} f(p)||^2$$
$$-||f(m) - \frac{1}{|\hat{A}(m)|} \sum_{a \in \hat{A}(m)} f(a)||^2 + \alpha, 0) \quad (8)$$

where $\alpha$ is the margin, $f(m)$ is the embeddings for the query mention $m$, $\frac{1}{|\hat{P}(m)|}\sum_{p\in\hat{P}(m)} f(p)$ is the mean of embeddings of the pseudo-positive labels $\hat{P}(m)$ and $\frac{1}{|\hat{A}(m)|}\sum_{a\in\hat{A}(m)} f(a)$ is the mean of embeddings of the pseudo-negative labels $\hat{A}(m)$.

The key intuition behind using the mean in a triplet loss formulation is to reduce overfitting to the noise in the pseudo labels. This works better in practice compared to the contrastive loss formulation in Equation (5) or mining a random positive/negative label for the standard triplet loss, especially when dealing with pseudo labels.

**(U2) Pseudo grounding loss (PGD)** Furthermore, we compute the pseudo grounding loss on the unlabeled training dataset. Specifically, we impute the pseudo-labels from the grounding function, $g(m,r)$. We only consider samples whose grounding score is greater than a confidence threshold $t$, which is set to 0.9 in our experiments. The high threshold value ensures that we consider only confident samples in the unlabeled set and eliminates learning from noisy samples. We denote this label after binary thresholding as $\hat{h}(m,r)$. The pseudo grounding alignment loss is:

$$\mathcal{L}_{pgd} = \sum_{m\in N}\sum_{r\in I} -\hat{h}(m,r)\log(g(m,r)) \quad (9)$$

Apart from the above mentioned task-based losses, we combine the standard image-text pretraining losses (Vaswani et al., 2017; Li et al., 2021). These losses help to learn better unimodal representations before fusion.

**(U3) Image-Text contrastive loss (ITC)** Following Goel et al. (2022), we incorporate the contrastive loss to align the image and narration pairs to learn better representations before fusion. This loss is defined as:

$$\mathcal{L}_{itc} = \sum_{m\in N} -\log\big(\frac{\exp(f_v(\boldsymbol{v}_r)f_t(m))}{\sum_{r'\in I}\exp(f_v(\boldsymbol{v}_{r'})f_t(m)))}\big)$$
$$(10)$$

where $f_v(\boldsymbol{v}_r)f_t(m)$ is the mention-region matching score from the visual and text representations before fusing in the multi-modal encoder and $\boldsymbol{v}_r$ are the raw features for the highest scoring region for a mention $m$.

**(U4) Masked language modeling loss (MLM)** To fine-tune the pretrained BERT model (Devlin et al., 2018) on the image-narration data, we also use the pre-trained task of masked language modeling. In particular, the input word tokens are randomly masked and are replaced by a special masking token. The model needs to predict the mask token based on the unmasked words. This task is trained with a cross-entropy loss, $\mathcal{L}_{mlm}$.

Hence, our overall training objective in Equation (4) is a combination of specialized task losses on the labeled training set $\mathcal{D}_s$ ($\mathcal{L}_{cr}$, $\mathcal{L}_{gd}$ and $\mathcal{L}_{bbr}$) and the unlabeled training set $\mathcal{D}_u$ ($\mathcal{L}_{pcr}$ and $\mathcal{L}_{pgd}$) and global pre-training objectives on the entire training dataset $\mathcal{D}$ ($\mathcal{L}_{itc}$ and $\mathcal{L}_{mlm}$).

### 3.4 Inference

To obtain the coreference scores, we form chains by measuring the cosine similarity between the mentions as described in Equation (1), considering the pairs with similarity higher than a predefined threshold as positives. When evaluating narrative grounding, we extract the cross-attention scores from the last layer of the multimodal encoder. For each mention, we identify the region with the highest softmax score as the positively referred region.

## 4 Experiments

**Datasets.** We evaluate our proposed method on the CIN dataset (Goel et al., 2022) that consists of 1000 test and 880 validation image-narration pairs from the Flickr30k split of the Localized Narratives dataset (Pont-Tuset et al., 2020) annotated with coreference chains and bounding boxes. We use the test split of the CIN dataset to report the performance on CR and narrative grounding. The annotations from the validation split are used as the small labeled set during training. The unlabeled dataset is the Flickr30k training subset of the Localized Narratives dataset, which consists of 50k image-narration pairs but is not annotated with bounding boxes or coreference chains.

**Implementation details.** For the image regions, we extract bounding box regions, visual features and object class labels using the Faster-RCNN object detector (Ren et al., 2015) as in Goel et al. (2022). We use a 4-layer transformer architecture for the text encoder and the multi-modal encoder similar to the ALBEF (Li et al., 2021) framework. The weights of the transformer encoders are initialized with the first four layers of BERT (Devlin et al., 2018). The visual encoder is a stack of two transformer encoder layers. Each transformer encoder layer includes a multi-head self-attention layer and an FFN. There are two heads in the multi-head attention layer, and two FC layers followed by ReLU

| Method | Modality | | MUC | | | $B^3$ | | | $CEAF_{\phi4}$ | | | CoNLL |
|---|---|---|---|---|---|---|---|---|---|---|---|---|
| | Text | Image | R | P | F1 | R | P | F1 | R | P | F1 | F1 |
| Neural Coref (Lee et al., 2017)[†*] | ✓ | ✗ | 0.11 | 0.17 | 0.13 | - | - | - | - | - | - | - |
| longdoc (Toshniwal et al., 2021)[*] | ✓ | ✗ | 7.79 | 8.43 | 7.24 | 62.27 | 76.10 | 67.69 | 48.77 | 84.95 | 61.02 | 45.31 |
| VisualBERT (Su et al., 2019)[*] | ✓ | ✓ | 18.17 | 6.08 | 8.06 | 69.01 | 36.08 | 41.03 | 21.25 | 57.10 | 28.67 | 25.92 |
| UNITER (Chen et al., 2020)[*] | ✓ | ✓ | 16.92 | 7.15 | 8.83 | 68.34 | 44.29 | 50.22 | 28.12 | 72.78 | 38.91 | 32.65 |
| VinVL (Zhang et al., 2021b)[*] | ✓ | ✓ | 16.76 | 8.60 | 9.75 | 68.49 | 62.32 | 61.30 | 42.88 | 80.81 | 53.69 | 41.58 |
| MAF (Wang et al., 2020) | ✓ | ✓ | 19.07 | 15.62 | 15.65 | - | - | - | - | - | - | - |
| WS-MCR (Goel et al., 2022) | ✓ | ✓ | 24.87 | 18.34 | 19.19 | - | - | - | - | - | - | - |
| Ours | ✓ | ✗ | 13.30 | 14.12 | 12.55 | 67.91 | 79.48 | 72.41 | 56.05 | 86.20 | 67.05 | 50.67 |
| | ✓ | ✓ | **31.11** | **35.25** | **31.86** | **70.63** | **87.85** | **78.06** | **63.99** | **93.44** | **75.47** | **61.79** |

Table 1: Coreference resolution results on the CIN dataset (Goel et al., 2022) from our proposed method and other state-of-the-art unimodal and multi-modal baselines. † indicates the use of predicted mentions, while the other results rely on ground-truth mentions during inference. ∗ means zero-shot performance.

activation layers in the FFN. Training details are in the appendix.

**Evaluation.** We report results for coreference resolution and narrative grounding. For the former, we use the standard CoNLL F1 score which is the average of three coreference-based metrics: MUC, $B^3$ and $CEAF_{\phi4}$. For the latter, we follow Goel et al. (2022) and report the grounding accuracy for both noun phrases and pronouns. More precisely, if the overlap between the ground-truth box and the predicted box is greater than 0.5, then it is considered to be a correct prediction.

## 5 Results and Discussion

### 5.1 Coreference Resolution

Table 1 reports the coreference resolution performance on the CIN dataset (Goel et al., 2022) for our method and the baselines. Further details about the baselines are given in the appendix. The text-based baselines Neural Coref (Lee et al., 2017) and longdoc (Toshniwal et al., 2021) are evaluated in a zero-shot way on the task. Their low CoNLL F1 scores indicate the incapability of the model to generalize to new domains which is in line with what has been evaluated extensively in the coreference literature (Toshniwal et al., 2021; Xia and Van Durme, 2021; Gandhi et al., 2023).

We further compare to strong multi-modal baselines by directly evaluating the VLMs in a zero-shot way on the CIN dataset. Interestingly, all three methods: VisualBERT (Su et al., 2019), UNITER (Chen et al., 2020) and VinVL (Zhang et al., 2021b) perform better in MUC and $B^3$ compared to the text-based baseline, longdoc (Toshni-

wal et al., 2021), but drop in performance on the average CoNLL F1 scores. These results show the inability of these models to effectively find singletons, hence leading to poor performance in the precision scores. Moreover, we can conclude that the vision and language pre-trained models fail to generalize for MCR.

We also compare to two weakly supervised methods that are trained on the CIN dataset, MAF (Wang et al., 2020) and WS-MCR (Goel et al., 2022). Goel et al. (2022) present results on the MAF model as a baseline and their proposed method, WS-MCR. MAF is a weakly supervised grounding method trained with ITC that is evaluated for CR and WS-MCR (Goel et al., 2022) learns weakly-supervised grounding and CR combining the ITC loss and prior linguistic rules. Both of these methods improve significantly in MUC scores compared to other zero-shot unimodal and multi-modal baselines.

Finally, we compare with the text-only variant (without any images) of our method. This method improves over the baselines on the CoNLL F1 scores. The significant gains in performance of our final method, with both text and image, combined with label supervision shows the importance of carefully tuning the model with a small amount of labeled data and large amounts of pseudo-labeled data.

### 5.2 Narrative Grounding

In Table 2, we present a comprehensive comparison between the baselines and our proposed approach on the task of narrative grounding. This task is both challenging and crucial, as it evaluates the precise

alignment between image regions and phrases in textual data. Notably, our proposed method goes beyond the traditional alignment of noun phrases and also addresses the grounding pronouns, which is vital for multimodal coreference resolution. We measure noun phrase grounding, pronoun grounding, and overall accuracy to measure performance (Goel et al., 2022).

Remarkably, our proposed method exhibits superior performance compared to weakly supervised baselines, showing a margin of improvement if approximately 2% and 2.5% in noun phrase and pronoun grounding accuracy, respectively. Furthermore, when compared to our unsupervised baseline, namely "Ours (ITC + MLM)", the inclusion of labeled and pseudo-labeled data yields a significant performance boost of approximately 6%. These results demonstrate the significance of training with both grounding alignment and coreference resolution loss, highlighting the mutual benefits derived from this approach.

| Method | Noun Phrases | Pronouns | Overall |
|---|---|---|---|
| MAF (Wang et al., 2020) | 21.60 | 18.31 | 20.91 |
| WS-MCR (Goel et al., 2022) | 30.27 | 25.96 | 29.36 |
| Ours (ITC + MLM) | 27.44 | 22.77 | 26.45 |
| Ours (Full) | **32.58** | **28.45** | **31.71** |

Table 2: Comparison of narrative grounding performance on the CIN dataset (Goel et al., 2022).

## 5.3 Ablation Study

**Varying labeled and unlabeled data.** We study the impact of labeled data on the learning process, allowing us to showcase the strengths of our approach. In Table 3, we measure the model's performance on CoNLL F1 scores at different proportions of labeled data (20% and 50%). Remarkably, despite the limited amount of labeled data samples, the model demonstrates consistently high performance without any significant drop. This highlights the exceptional ability of our model to effectively learn from a small labeled set, without relying on a large number of annotated training samples.

Furthermore, to validate the efficacy of our proposed method, we also investigate the influence of unlabeled data samples during training. Following the same data split as in the supervised experiments, we observe the changes in performance indicated by row 2 in Table 3. As the quantity of unlabeled samples increases, the model exhibits enhanced coreference resolution performance. This result

| Data type | % Samples | CoNLL F1 |
|---|---|---|
| Labeled | 20% | 60.04 |
| | 50% | 61.24 |
| Unlabeled | 20% | 56.82 |
| | 50% | 59.11 |

Table 3: CR performance by changing the amount of labeled and unlabeled data during training.

reinforces the ability of our proposed method to leverage and effectively learn from pseudo-labeled data. Detailed results are in the appendix.

**Impact of different loss functions.** In Table 4, we assess the performance of coreference resolution by incorporating various losses proposed in Section 3. Throughout the training process, the model consistently integrates the self-supervised objectives of ITC and MLM, see first row of Table 4.

Integrating the supervised contrastive coreference resolution loss, CR, in addition to ITC and MLM, results in a significant performance drop. Due to the limited availability of labeled data, the model struggles to effectively generalize for coreference resolution, leading to overfitting and consequently lower F1 scores. However, by progressively incorporating the bounding box regression loss, BBR, and the grounding alignment loss GD, we get a much stronger training signal even with a small labeled set. This multi-task training objective contributes to an improvement of approximately 1.5% in the CoNLL F1 score.

Subsequently, we investigate the impact of incorporating loss on pseudo-labeled data. By introducing the pseudo coreference loss, denoted as PCR, we observe a remarkable improvement of approximately 2% in the CoNLL F1 scores. This result highlights the significance of leveraging pseudo clusters and underscores the effectiveness of our proposed robust triplet loss, which computes the triplet loss using the mean of positive and negative embeddings. Notably, this approach successfully incorporates pseudo-labeled data without leading to overfitting while achieving substantial performance gains. Consequently, our final proposed method, which integrates the pseudo grounding loss, PGD, exhibits the most superior overall performance, validating the potency of pseudo-labels for both coreference resolution and grounding.

**Choice of coreference resolution loss.** In Table 5, we examine the impact of different types of coreference resolution losses. We present a comparison of the following loss combinations: (1) Binary

| Losses | | | | | | | MUC | | | $B^3$ | | | CEAF$_{\phi4}$ | | | CoNLL |
|---|---|---|---|---|---|---|---|---|---|---|---|---|---|---|---|---|
| PCR(U1) | PGD(U2) | ITC(U3) | MLM(U4) | CR (S1) | GD (S2) | BBR (S3) | R | P | F1 | R | P | F1 | R | P | F1 | F1 |
| ✗ | ✗ | ✓ | ✓ | ✗ | ✗ | ✗ | 23.81 | 25.83 | 23.12 | 69.32 | 85.87 | 76.41 | 61.00 | 89.69 | 72.05 | 57.19 |
| ✗ | ✗ | ✓ | ✓ | ✓ | ✗ | ✗ | 22.70 | 21.40 | 20.23 | 69.05 | 80.03 | 73.66 | 55.52 | 87.09 | 67.20 | 53.70 |
| | | | | ✓ | ✗ | ✓ | 23.86 | 24.52 | 22.31 | 69.31 | 84.15 | 75.67 | 59.50 | 89.15 | 70.80 | 56.26 |
| | | | | ✓ | ✓ | ✓ | 27.68 | 29.04 | 26.66 | 69.93 | 85.43 | 76.61 | 60.92 | 90.61 | 72.26 | 58.51 |
| ✓ | ✗ | ✓ | ✓ | ✓ | ✓ | ✓ | 30.66 | 32.82 | 30.31 | 70.70 | 86.09 | 77.33 | 62.64 | 92.92 | 74.27 | 60.64 |
| ✓ | ✓ | | | | | | **31.11** | **35.25** | **31.86** | **70.63** | **87.85** | **78.06** | **63.99** | **93.44** | **75.47** | **61.79** |

Table 4: Ablation study on our proposed method with the combination of the proposed losses.

| CR Loss | | MUC | | | $B^3$ | | | CEAF$_{\phi4}$ | | | CoNLL |
|---|---|---|---|---|---|---|---|---|---|---|---|
| on $\mathcal{D}_s$ | on $\mathcal{D}_u$ | R | P | F1 | R | P | F1 | R | P | F1 | F1 |
| BCE | BCE | 23.63 | 23.55 | 21.57 | 69.50 | 81.94 | 74.47 | 57.68 | 88.01 | 68.95 | 55.00 |
| CR | CR | 28.20 | 22.47 | 23.08 | 70.10 | 76.29 | 72.40 | 52.32 | 87.22 | 64.71 | 53.40 |
| CR | RTC | 29.24 | 32.41 | 29.46 | 70.37 | 86.71 | 77.45 | 63.14 | 92.59 | 74.55 | 60.49 |
| CR | PCR | **31.11** | **35.25** | **31.86** | **70.63** | **87.85** | **78.06** | **63.99** | **93.44** | **75.47** | **61.79** |

Table 5: Performance comparison with the choice of coreference resolution loss on the labeled dataset $\mathcal{D}_s$ and the unlabeled dataset $\mathcal{D}_u$.

cross-entropy loss (BCE) applied to both $\mathcal{D}_s$ and $\mathcal{D}_u$, (2) Supervised contrastive loss (CR) applied to both $\mathcal{D}_s$ and $\mathcal{D}_u$, and (3) Supervised contrastive loss (CR) on $\mathcal{D}_s$ and random triplet mining loss (RTC) on $\mathcal{D}_u$.

We observed a significant performance drop when training with the BCE loss, compared to utilizing the supervised contrastive loss. We hypothesize that the supervised contrastive loss provides a better clustering of mentions by contrasting them in the embedding space directly than the binary cross-entropy loss. Consequently, the embeddings become more robust for CR, contributing to improved performance.

Interestingly, when applying the supervised contrastive loss to $\mathcal{D}_u$ (row 2), we observed a drop in performance. Our hypothesis is that the contrastive loss tends to overfit in the presence of noisy pseudo labels, leading to a degradation in performance. In contrast, our pseudo triplet loss formulation PCR is softer in penalizing noisy pseudo labels. This allows the model to gradually adapt and become more resilient to such noise, resulting in more efficient clustering of mentions. We also compare to another ablation where instead of taking the mean of the embeddings for pseudo-positive labels and pseudo-negative labels, we sample a random positive and negative label (results in row 3) abbreviated as RTC. Randomly sampling the labels generalizes better than the other ablations but the mean cluster embeddings outperforms than randomly selecting samples.

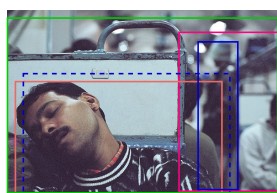

WS-MCR (Goel et al., 2022)

in this image i can see **people** sitting in **train** and the front man is sleeping and a man standing and the background is blurry.

**Ours**

in this image i can see **people** sitting in **train** and **the front man** is sleeping and **a man** standing and the background is blurry.

Figure 3: Visualization for grounding and coreference resolution. The colored boxes in image correspond to the mentions with the same color in the sentence.

### 5.4 Qualitative Results

In Figure 3, we qualitatively visualize the performance of our method and compare it with the weakly supervised baseline from Goel et al. (2022). Our model correctly separates the mentions *the front man* and the *the man* both during CR and grounding, whereas the WS-MCR (Goel et al., 2022) method incorrectly assigns the mention *the man* to the *the front man* and grounds it incorrectly too (denoted by the blue dotted line). Hence, our method can effectively learn to disambiguate the instances based on the visual details which is also helpful for coreference resolution.

### 6 Conclusion

In conclusion, this paper addresses the challenging task of multimodal coreference resolution where an image is accompanied by a longer descriptive text. We propose a data efficient semi-supervised approach that incorporates task-based losses for both labeled and unlabeled data, operating within a cross-modal framework. Our method achieves remarkable results for CR and narrative grounding tasks on the CIN dataset, showcasing its effectiveness in handling the complexities of MCR. In the future, we plan to investigate how the power of pre-training combined with semi-supervised fine-tuning can be fully utilized for the task of MCR.

## Limitations

Here, we outline limitations that are important considerations for future work.

First, the current model's performance in coreference resolution and grounding is limited by the use of a pre-trained object detector. The detectors pretrained for object detection task have a limited object category vocabulary and lack in fine-grained properties including adjectives, human actions and the open vocabulary found in narrations. This forces the model to rely on a predetermined set of regions and object classes, preventing it from directly learning region coordinates for a mention on an image. To improve performance, we envision the development of an end-to-end approach that eliminates this reliance on pre-defined regions.

Second, our model currently depends on ground-truth mentions to resolve coreferences and ground them. In the future, one promising direction would be to detect mentions simultaneously with coreference resolution and grounding. This would significantly improve the applicability of our proposed method and reduce dependence on off-the-shelf mention detectors or ground-truth annotations.

## Ethics Statement

All datasets used in this work have been previously released. The use of the CIN dataset (Goel et al., 2022) in our paper is consistent with their intended use. The detail of the dataset is describied in Goel et al. (2022). Multimodal datasets frequently include social biases, and we expect the models trained on them to reflect the biases in these datasets. It is important to note that multimodal models have both beneficial and harmful applications. Beneficial applications include advanced image and video retrieval, visual description systems to assist the visually impaired, and user interfaces that enhance interaction with smart home devices. However, harmful applications, such as non-consensual surveillance or fine-tuning models to retrieve inappropriate content, must be carefully addressed and mitigated.

## Acknowledgements

AG is supported by the Armeane Choksi Scholarship and HB is supported by the EPSRC programme grant Visual AI EP/T028572/1 and HB and FK are supported by Edinburgh Laboratory for Integrated Artificial Intelligence (ELIAI).

This research/project is supported by the National Research Foundation, Singapore, under its NRF Fellowship (Award NRF-NRFF14-2022-0001). Thanks to Nikita Moghe and Matt Grenander for discussions and constructive feedback.

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

# Appendix

## A  Baselines

We consider the following baselines to fairly compare and evaluate our proposed method:
**(a) Text-only CR:** For all these methods, we directly evaluate the coreference chains using the narration only without the image. (1) *Neural-Coref (Lee et al., 2017):* This method is trained end-to-end using a neural network on a large corpus of wikipedia data to detect mentions and coreferences. For this baseline, we use the predicted mentions instead of the gold mentions. (2) *longdoc (Toshniwal et al., 2021):* This is a strong transformer based method using Longformer-Large as the backbone for coreference resolution, trained on multiple datasets. We use the gold mentions to predict coreference chains for this model.
**(b) Multi-modal CR:** We evaluate strong vision and langauge models for the task on coreference resolution on the CIN dataset. (1) *VisualBERT (Su et al., 2019), UNITER (Chen et al., 2020), VinVL (Zhang et al., 2021b):* All these three baselines are strong vision language models trained on image-caption data and shows improvements on a variety of downstream tasks such as VQA, NLVR etc. To test it for CR, we compute the cosine similarity for the multi-modal mention embeddings in a zero-shot way. (2) *MAF (Wang et al., 2020):* MAF is a weakly supervised phrase grounding method, originally trained on the Flickr30k-Entities (Plummer et al., 2015). Goel et al. (2022) train this model on narrations data and evaluate it for CR. (3) *WS-MCR (Goel et al., 2022):* This is a strong weakly supervised method for multimodal coreference resolution on the CIN dataset (Goel et al., 2022). In this method, they train a vision and text encoder with an image-text contrastive loss and weak prior linguistic rules as a regularizer.

## B  Bounding box regression loss (BBR)

We define the smmoth-L1 loss (Ren et al., 2015) for the bounding box transformation as follows:

$$\mathcal{L}_{bbr} = \begin{cases} 0.5(b_m - b'_m)^2/\beta, & \text{if } |b_m - b'_m| < \beta \\ |b_m - b'_m| - 0.5 * \beta, & \text{otherwise} \end{cases}$$

(11)

where $\beta$ is set to 1 following previous work (Girshick, 2015; Ren et al., 2015), $b'_m$ are the transformed bounding box coordinates after applying the delta transformation $\delta_m$ on the maximum region proposal $r_m$ similar to (Girshick, 2015).

## C  Training details

The whole architecture is trained end-to-end with the AdamW (Loshchilov and Hutter, 2017) optimizer. The initial learning rate of the model is 1e-5. The learning rate is gradually warmed up for 2 epochs with a unit multiplier and then decayed following a step scheduler with step size of 10 epochs and gamma of 0.95. We use a batch size of eight and weight decay of 0.01. The model is trained for 30 epochs and we choose the best performing model based on the test set. The model is trained on 4 V100 GPUs with data parallelism. All code and models will be made available at https://github.com/VICO-UoE/CIN-SSL.

## D  Further Ablations

**Pre-trained weights from ALBEF (Li et al., 2021).** In Table 6, we show CR results by replacing the text-encoder and the multi-modal encoder in our model from the ALBEF (Li et al., 2021) framework with 6 transformer blocks in each encoders. Moreover, we use the pre-traiend weights from ALBEF (Li et al., 2021) and then fine-tune using our semi-supervised training strategies. Despite strong pre-training from ALBEF, the model does not fine-tune and transfer well to the task of MCR compared to our method which is initialized with BERT weights.

**Varying the threshold for pseudo-grounding loss.** In Table 7, we compare the results with different grounding thresholds for the pseudo grounding loss. Including all the pseudo predictions (threshold of 0.0) leads to a significant drop in performance showing the importance of thresholding-based training strategy. Furthermore, as presented in the results the threshold of 0.9 works the best compared to 0.5 and 0.7.

**Varying the amount of labeled and unlabeled data.** In Table 8 and Table 9, we present detailed results from the discussion in Section 5.3 on different CR metrics by varying the amount of labeled and unlabeled data.

## E  Qualitative Results

In Figure 4 and Figure 5, we present detailed qualitative visualizations. Figure 4 shows the coreference chains predicted by our proposed method and the baseline, WS-MCR (Goel et al., 2022). The baseline model misses the instances of *his* to relate it to the *the man* (row 2, column 2) which is

| Method | MUC | | | B$^3$ | | | CEAF$_{\phi4}$ | | | CoNLL |
| --- | --- | --- | --- | --- | --- | --- | --- | --- | --- | --- |
| | R | P | F1 | R | P | F1 | R | P | F1 | F1 |
| Weights from ALBEF (Li et al., 2021) | **31.70** | 25.97 | 26.47 | 70.78 | 78.35 | 73.77 | 54.58 | 89.02 | 66.94 | 55.73 |
| Ours (Weights from BERT (Devlin et al., 2018)) | 31.11 | **35.25** | **31.86** | **70.63** | **87.85** | **78.06** | **63.99** | **93.44** | **75.47** | **61.79** |

Table 6: CR performance with ALBEF (Li et al., 2021) as pre-trained weights.

| Grounding threshold | MUC | | | B$^3$ | | | CEAF$_{\phi4}$ | | | CoNLL |
| --- | --- | --- | --- | --- | --- | --- | --- | --- | --- | --- |
| | R | P | F1 | R | P | F1 | R | P | F1 | F1 |
| 0.0 | 28.20 | 22.47 | 23.08 | 70.10 | 76.29 | 72.40 | 52.32 | 87.22 | 64.71 | 53.40 |
| 0.5 | 31.18 | 30.04 | 28.92 | 70.80 | 82.67 | 75.76 | 59.68 | 92.18 | 71.77 | 58.82 |
| 0.7 | 30.48 | 33.34 | 30.58 | 70.63 | 86.74 | 77.58 | 63.30 | 93.22 | 74.85 | 61.01 |
| 0.9 | **31.11** | **35.25** | **31.86** | **70.63** | **87.85** | **78.06** | **63.99** | **93.44** | **75.47** | **61.79** |

Table 7: Performance of our proposed method by varying the grounding threshold $t$ to include samples above this threshold in Equation (9).

| % $\mathcal{D}_s$ | MUC | | | B$^3$ | | | CEAF$_{\phi4}$ | | | CoNLL |
| --- | --- | --- | --- | --- | --- | --- | --- | --- | --- | --- |
| | R | P | F1 | R | P | F1 | R | P | F1 | F1 |
| 20% | 26.40 | 31.18 | 27.26 | 69.83 | 88.26 | 77.75 | 64.25 | 92.02 | 75.11 | 60.04 |
| 50% | 28.65 | 34.13 | 29.91 | 70.26 | 88.83 | 78.27 | 64.55 | 92.56 | 75.53 | 61.24 |
| 100% | **31.11** | **35.25** | **31.86** | **70.63** | **87.85** | **78.06** | **63.99** | **93.44** | **75.47** | **61.79** |

Table 8: CR performance by varying the number of labels in the labeled dataset.

| % $\mathcal{D}_u$ | MUC | | | B$^3$ | | | CEAF$_{\phi4}$ | | | CoNLL |
| --- | --- | --- | --- | --- | --- | --- | --- | --- | --- | --- |
| | R | P | F1 | R | P | F1 | R | P | F1 | F1 |
| 20% | 30.96 | 27.09 | 27.20 | 70.84 | 79.36 | 74.23 | 56.49 | 90.79 | 69.04 | 56.82 |
| 50% | 30.87 | 29.97 | 28.83 | 70.75 | 83.58 | 76.23 | 60.24 | 92.02 | 72.27 | 59.11 |
| 100% | **31.11** | **35.25** | **31.86** | **70.63** | **87.85** | **78.06** | **63.99** | **93.44** | **75.47** | **61.79** |

Table 9: CR performance by varying the number of labels in the unlabeled dataset.

2022). Compared to the baseline, our method successfully grounds *entrance door*, *some other people* and *two women*. Hence our method clearly exhibits strong coreference resolution and grounding capabilities compared to previous work.

correctly clustered by our method. Moreover, WS-MCR (Goel et al., 2022) cannot relate *big orange color building* to *the building* (row2, column 4) unlike our method. This highlights that our method is able to learn fine-grained correlations between the image regions and text to effectively resolve such ambiguities. Despite significant advantages, our method still fails to resolve cases like *some other people* by clustering them into the same chain. We believe that the model needs more complex visual understanding (localize different instance of *some other people*) and contextual knowledge from text (*people standing on road* vs *people standing on footpath*) for these specific cases.

In Figure 5, we show grounding of the corresponding mentions on the image from our proposed method and the baseline WS-MCR (Goel et al.,

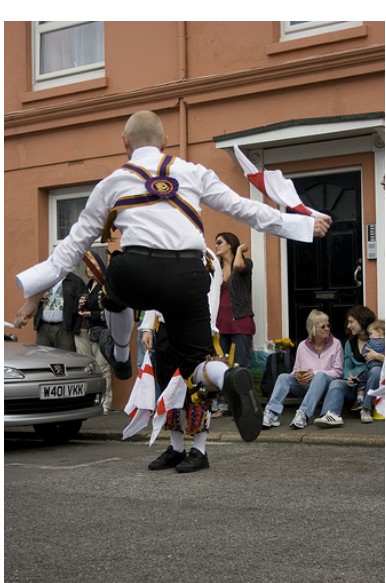

Ground Truth

in this image we can see **a man** is dancing on **a road**, and **the man** is wearing **white color shirt** and **black color pant.**

we can also see a **red and white color cloth** in **his hand**, and **the man** is also wearing a **orange and black color belt** around **his** chest

on **the road** we can see a **grey color car** is parked, and we can also see **some other people** also standing on **the road** by holding **flags**

on **the footpath** we can see **two women** are sitting and talking with each other, and in **that one woman** is holding **a baby,** and we can also see **some other people** are also standing on **the footpath**, and in the background we can see **a big orange color building** with **white color glass windows**, and we can also see the entrance door of **the building**.

WS-MCR (Goel et al., 2022)

in this image we can see **a man** is dancing on **a road**, and **the man** is wearing **white color shirt** and **black color pant.**

we can also see a **red and white color cloth** in **his hand**, and **the man** is also wearing a **orange and black color belt** around **his** chest

on **the road** we can see a **grey color car** is parked, and we can also see **some other people** also standing on **the road** by holding **flags**

on **the footpath** we can see **two women** are sitting and talking with each other, and in **that one woman** is holding **a baby,** and we can also see **some other people** are also standing on **the footpath**, and in the background we can see **a big orange color building** with **white color glass windows**, and we can also see the entrance door of **the building.**

Ours

in this image we can see **a man** is dancing on **a road**, and **the man** is wearing **white color shirt** and **black color pant.**

we can also see a **red and white color cloth** in **his hand**, and **the man** is also wearing a **orange and black color belt** around **his** chest

on **the road** we can see a **grey color car** is parked, and we can also see **some other people** also standing on **the road** by holding **flags**

on **the footpath** we can see **two women** are sitting and talking with each other, and in **that one woman** is holding **a baby,** and we can also see **some other people** are also standing on **the footpath**, and in the background we can see **a big orange color building** with **white color glass windows**, and we can also see the entrance door of **the building**.

Figure 4: Coreference resolution for an image-narration pair. We break down the full narration as individual sentences in the columns for simplicity. The rows from top to bottom show ground-truth annotations, predictions from the WS-MCR method (Goel et al., 2022) and Ours. The mentions in the same color form a part of the coreference chain.

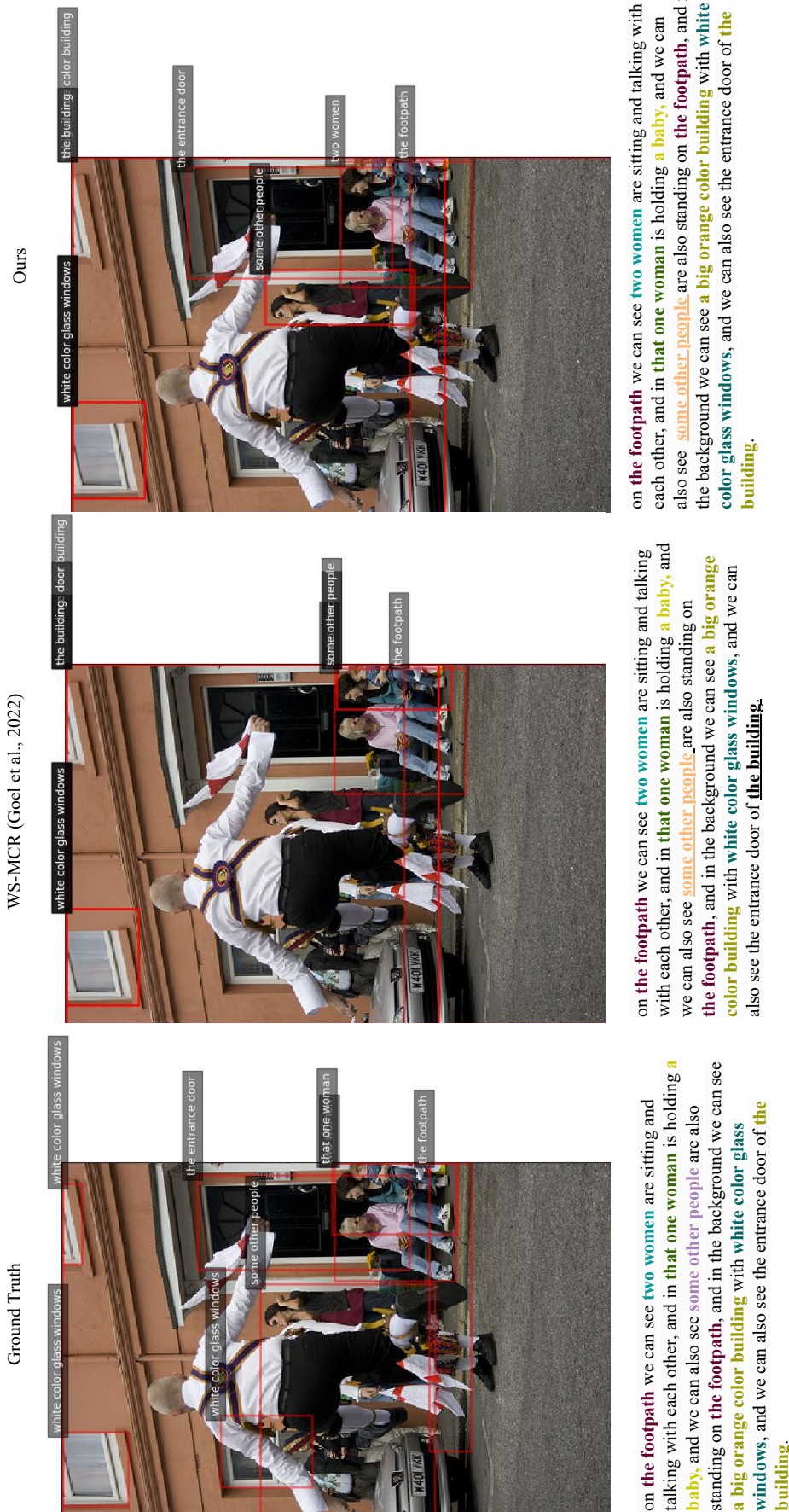

Figure 5: Visualization of Grounding and CR performance. Zoom in for better visualization of bounding boxes.