# OpenReview forum: "Semi-supervised multimodal coreference resolution in image narrations"
_EMNLP/2023/Conference — EMNLP 2023 Main_

### Official Review · Reviewer_B6pm · 2023-07-31

**Soundness:** 4

**Excitement:**

4: Strong: This paper deepens the understanding of some phenomenon or lowers the barriers to an existing research direction.

**Paper Topic And Main Contributions:**

The paper introduces a semi-supervised cross-modal coreference resolution model. In addition to linking mentions in a text, the mention clusters are also connected to regions in images that are described in the respective text. The training makes use of a variety of loss functions some supervised, some unsupervised. The presented approach archives state-of-the-art scores for the task on the CIN dataset.

**Questions For The Authors:**

A) What is the reason for using gold data for text mentions but extracting the image regions automatically? Does the image region depend on the gold data or do predicted bounding boxes also work? In general, I think the usage of gold mentions needs to be made explicit early in the text.

B) Is there a reason you did not use LEA as an evaluation metric? While it is not as popular, it is in theory much superior to the other measures, hence I would love to see it used more.

C) Is U1 only calculated for the text narrations?

D) U1: What exactly does h_m refer to?

E) U1: How are the mentions for this loss generated, just any random text span?

F) U1: How is the m embedding for initial positive/negative sampling generated?

G) Line 395 onwards, does this imply a coreference chain can be associated with multiple image regions?

H) Previous work has shown that even just training on domain-specific mentions can yield large improvements (https://aclanthology.org/2023.acl-long.588/). Wouldn't it make sense to test existing coreference models not just in a zero-shot setting but also in some sort of few-shot setup? (This should not be considered a missing reference, as it is a very recent publication)

I)  "Moreover, it validates
453 the need for multi-modal models for the task of
454 multimodal coreference resolution." Why is this the case? The low f1 to me just indicates that models are very domain specific.

**Reasons To Accept:**

The paper produces state-of-the-art results on a widely applicable task, it is relevant to a wide range of multimodal tasks. It shows clear improvements over reported SotA results. The paper shows a performance increase when adding the image and attributes this to the poor singleton identification of text-only models. Qualitative evaluation is used to showcase and explain the models' strengths.

**Reasons To Reject:**

Some of the architecture choices are not motivated well, and relatively little investigation into the reason for the positive impact of individual loss functions is provided. This is a problem as improvements brought about by the loss functions are the paper's main contribution.
Perhaps this is a lack of experience on my side as I am not very familiar with multi-modal approaches, but the PCR loss does not make sense to me despite the given intuitive explanation. From my understanding, it just incorporates the models existing understanding of distance into the training. The paper clearly shows it still performing well but lacks an exploration of why. My question would be if it essentially acts as encoding a cosine-distance based coreference resolution policy. A few minor changes in the paper may potentially be able to clear this up (see questions).

The authors do not validate their text coreference against very recent models e.g. the word-level coreference model, with the Toshniwal et al model specifically targeting long documents at the expense of accuracy.  As such the coreference approach explored here seems to effectively ignore a wide body of coreference work. To be fair this seems to also be the case for the CIN dataset paper (Goel et al, 2022).

It is not clear at first sight which methods are trained and which are evaluated in a zero-shot setting. While this is easy to fix I believe that, in the current state, the results are slightly misleading.

For me, there were a few cases in which lack of detail impacted my understanding (e.g. PCR loss but also 395-397)

**Reproducibility:**

4: Could mostly reproduce the results, but there may be some variation because of sample variance or minor variations in their interpretation of the protocol or method.

**Reviewer Confidence:**

3: Pretty sure, but there's a chance I missed something. Although I have a good feel for this area in general, I did not carefully check the paper's details, e.g., the math, experimental design, or novelty.

**Typos Grammar Style And Presentation Improvements:**

Please consider using booktabs for your tables (especially using \cmidrule for grouping columns, e.g. P, R, and F1 for any of the metrics), the interrupted vertical lines are a bit visually distracting for me, please consider removing them. In my opinion, this would help the visual presentation of the paper immensely.

345: is "impute" the correct verb here?

"Table" as in "Table 3" (e.g. line 535) are typically capitalized, same for section.

Please add a label indicating unsupervised losses in Table 4, it is hard to keep the abbreviations in mind at all times.

---

> ### Author Rebuttal · Authors · 2023-08-28
>
> We thank the reviewer for the valuable comments which we respond to below.
>
> > Some of the architecture choices are not motivated well, and relatively little investigation into the reason for the positive impact of individual loss functions is provided. This is a problem as improvements brought about by the loss functions are the paper's main contribution.
>
>
> We would like to emphasize that we comprehensively evaluate the importance of adding each loss term in Table 4, the choice of loss for coreference resolution in Table 5 and a discussion about the importance of losses in Section 5.3. Moroever, we also study the impact of adding unlabeled data during training in Table 3.
>
>
>
>
> > Perhaps this is a lack of experience on my side as I am not very familiar with multi-modal approaches, but the PCR loss does not make sense to me despite the given intuitive explanation. From my understanding, it just incorporates the models existing understanding of distance into the training. The paper clearly shows it still performing well but lacks an exploration of why. My question would be if it essentially acts as encoding a cosine-distance based coreference resolution policy. A few minor changes in the paper may potentially be able to clear this up (see questions).
>
>
> The PCR loss computes the coreference loss over the unlabeled data. To compute this loss, the model treats its own coreference predictions as pseudo-labels. Hence, to prevent overfitting of the model to the wrong predictions, at each iteration, we only consider highly confident positive and negative predictions (coreference links) from the model in a triplet loss formulation (Eq. 8). This process goes through several iterations, each time adding updated pseudo-labels hence improving the performance of the model. This self-learning paradigm has been shown to be extremely effective in literature as the selection of unlabeled samples with high-confidence predictions moves decision boundaries to low-density regions [1,2].  We will revise the explanation in the paper.
>
> [1]Oymak, Samet, and Talha Cihad Gulcu. "Statistical and algorithmic insights for semi-supervised learning with self-training." arXiv preprint arXiv:2006.11006 (2020).
>
> [2]Rizve, Mamshad Nayeem, Kevin Duarte, Yogesh S. Rawat, and Mubarak Shah. "In defense of pseudo-labeling: An uncertainty-aware pseudo-label selection framework for semi-supervised learning." arXiv preprint arXiv:2101.06329 (2021).
>
>
> > The authors do not validate their text coreference against very recent models e.g. the word-level coreference model, with the Toshniwal et al model specifically targeting long documents at the expense of accuracy. As such the coreference approach explored here seems to effectively ignore a wide body of coreference work. To be fair this seems to also be the case for the CIN dataset paper (Goel et al, 2022).
>
>
> As suggested, we now provide a new neural coref baseline for a fair comparison that fine-tunes the Longformer-large model (Toshniwal et al. 2021) only on the labeled set of the CIN dataset. Its weights are initialized with the pre-trained model on other known 8 coref datasets including OntoNotes, PreCo. Surprisingly, it is nontrivial to finetune this large model without overfitting into the small labeled set of the CIN. The CoNLL F1 score for the fine-tuned model is 42.41 which is slightly lower than the score of 45.31 from the pre-trained model (Table 1, 2nd row). This also emphasizes the need for learning from larger unlabeled data. We will add these results to the paper and discuss them.
>
>
> > It is not clear at first sight which methods are trained and which are evaluated in a zero-shot setting. While this is easy to fix I believe that, in the current state, the results are slightly misleading.
>
>
> Thanks for pointing this out. We will address this in the final version.
>
>
> > What is the reason for using gold data for text mentions but extracting the image regions automatically? Does the image region depend on the gold data or do predicted bounding boxes also work? In general, I think the usage of gold mentions needs to be made explicit early in the text.
>
>
> Along with coreference resolution, our objective also involves grounding i.e. predicting a bounding box for a mention. To tackle this, we adopt a two-stage approach. In the first stage, we employ an object detector to extract bounding boxes. Subsequently, the second stage involves learning the matching process with respect to these extracted boxes. These predicted bounding boxes act as inputs during both the training and inference phases. While directly using the ground-truth boxes is possible it is not scalable as they are expensive to obtain, are missing for the unlabeled part of the dataset, and is orthogonal to the objective of learning grounding. In the future, as an extension, we plan to jointly learn the bounding box prediction for mentions along with coref.
>
> For coreference resolution, during the training process, we utilize mentions extracted from an off-the-shelf NLP parser (https://spacy.io/) as outlined in Goel et al. (2022). However, during the inference stage, we opt for gold mentions, aligning with established practices in previous coreference research. As outlined in the limitations, we will extend our work to predict mentions along with coreference chains. We will ensure that these distinctions are explicitly clarified in the text.
>
>
> > Is there a reason you did not use LEA as an evaluation metric? While it is not as popular, it is in theory much superior to the other measures, hence I would love to see it used more.
>
>
> Thanks for the suggestion. We use the evaluation metrics commonly used in the coref literature and as proposed in the original work. As suggested, we present new results on the LEA metric for both the supervised and semi-supervised setup. The results are in the table below:
>
>
> | Ours | CoNLL F1 |  LEA-P  |  LEA-R  |  LEA-F1  |
> | --- | ----------- |  ---  |  ---  |  ---  |
> | Supervised | 58.51 |  42.07  |  60.66  |  49.68  |
> | Final | 61.79 |  58.72  |  70.72  |  64.17  |
>
>
> Our final semi-supervised method surpasses in the LEA metric by significant margins compared to the carefully designed supervised training. We will add these results in the final version.
>
>
> > Is U1 only calculated for the text narrations?
>
> U1 is only calculated for text mentions, where the embeddings for the mentions are the output from the multimodal encoder as given in Eq 3.
>
>
> > U1: What exactly does h_m refer to?
>
> Thanks for pointing this out. It is indeed a typo and h_m should be f(m) which is the embeddings from the multi-modal encoder for the query mention m.
>
>
> > U1: How are the mentions for this loss generated, just any random text span?
>
> During training, we use the mentions generated from an off-the-shelf NLP parser (https://spacy.io/) following Goel et al. (2022). We will make it clear in the implementation details.
>
>
> > U1: How is the m embedding for initial positive/negative sampling generated?
>
> The embeddings for the positive and negative mentions are generated using the output from the multimodal encoder as given in Eq 3, similar to the query embedding f(m).
>
>
> > Line 395 onwards, does this imply a coreference chain can be associated with multiple image regions?
>
> A coreference chain always corresponds to a single image region except where there are multiple bounding boxes for the mentions, eg. “three men”, in which case there could be three bounding boxes. We will clarify this in the text.
>
>
> > Previous work has shown that even just training on domain-specific mentions can yield large improvements (https://aclanthology.org/2023.acl-long.588/). Wouldn't it make sense to test existing coreference models not just in a zero-shot setting but also in some sort of few-shot setup? (This should not be considered a missing reference, as it is a very recent publication)
>
> Thanks for the reference. Due to the short rebuttal period, we were unable to setup an experiment for the given paper. This is however an interesting direction and we will consider this in future work as this might lead to further improvements when combined with our training regime.
>
>
> > "Moreover, it validates 453 the need for multi-modal models for the task of 454 multimodal coreference resolution." Why is this the case? The low f1 to me just indicates that models are very domain specific.
>
> We agree that the low F1 is due to the domain specificity of the models. However, we argue that these models require a large labeled dataset to close the domain gap as also corroborated by our new experiment on Longformer-large (as discussed above). As shown in the experiments adding text and image modalities (Table 1, last row) in a semi-supervised framework mitigates this issue in a low data regime, especially when compared to the model with only text modality (Table 1, 2nd last row). Hence, we conclude that our model can be efficiently trained on the combined labeled and unlabeled data in a multi-modal setup.
>
>
> > Grammar and presentation improvements
>
> Thanks for the suggestions, we will update these.

---

### Official Review · Reviewer_X5uF · 2023-08-05

**Typos Grammar Style And Presentation Improvements:** N/A
**Soundness:** 4

**Excitement:**

4: Strong: This paper deepens the understanding of some phenomenon or lowers the barriers to an existing research direction.

**Missing References:**

N/A

**Paper Topic And Main Contributions:**

In this paper, the authors proposed a framework that tackles coreference resolution across vision (image) and text (narration or caption) as well as within text. This is a semi-supervised framework. The images and texts are fed into visual encoder and text encoder respectively. The text embeddings are fed into a self-attention module which is followed by a cross attention layer that consumes visual embeddings. With an additional feed forward layer, the attention modules form a multimodal encoder. The MM encoders are trained through self learning technique for masked language model (MLM) and image-text contrastive loss. Moreover, with the information from labeled data, the MM encoders are updated with coreference loss, grounding loss and bounding box regression loss; while the unlabeled data provides pseudo coreference loss and pseudo grounding loss. The authors also conduct excessive experiments to compare against the SOTA baselines as well as the ablation settings with varied labeled and unlabeled data and different combinations of loss functions.

**Questions For The Authors:**

N/A

**Reasons To Accept:**

This paper has the following merits:
1. The paper is solid in terms of methodology. The framework is convincing in theory and the experiments also support their claim.
2.  It is easy to follow, researchers can implement with scratch by simply reading the paper and the instructions in appendix.
3. The authors carefully present their experiment sections with numbers, analysis and qualitative examples.

**Reasons To Reject:**

I don't have any particular reasons to reject the paper as I am leaning to give a thumb-up for this paper.

**Reproducibility:**

5: Could easily reproduce the results.

**Reviewer Confidence:**

4: Quite sure. I tried to check the important points carefully. It's unlikely, though conceivable, that I missed something that should affect my ratings.

---

> ### Author Rebuttal · Authors · 2023-08-28
>
> We thank the reviewer for their feedback and acknowledgment of the effectiveness of our method, clear writing, and extensive experiments. We will address any further concerns that may be raised during the discussion period.

---

### Official Review · Reviewer_xjtT · 2023-08-05

**Soundness:** 4

**Excitement:**

4: Strong: This paper deepens the understanding of some phenomenon or lowers the barriers to an existing research direction.

**Missing References:**

L452: Yang et al., 2022 is more focused on coref in LLMs, which doesn’t seem like the right citation for the model scale in this work/longdoc;
[Moving on from OntoNotes: Coreference Resolution Model Transfer](https://aclanthology.org/2021.emnlp-main.425) (Xia & Van Durme, EMNLP 2021) and [Annotating Mentions Alone Enables Efficient Domain Adaptation for Coreference Resolution](https://aclanthology.org/2023.acl-long.588) (Gandhi et al., ACL 2023) would be better citations if you want more.

**Paper Topic And Main Contributions:**

The paper proposes a model for visual coreference given image and narrations. Their semi-supervised model uses a fused text-and-vision transformer with multiple supervised and unsupervised objectives so that they can train on a small set of labeled data and a much larger set of unlabeled data. They demonstrate the effectiveness of this model on the Coreferenced Image Narratives (CIN) dataset for coreference resolution (of the narrations) and narrative grounding (match text cluster to image). Across multiple ablations, they show the effect of each of the training objectives and the pseudo-labels that are created for the unsupervised objectives.

**Questions For The Authors:**

* Why doesn’t the grounding model use the ground-truth bounding boxes during training, or at least parts of training (L287-289) for g? This could be viewed as some kind of teacher forcing and should be an objective if it is available?
* L421: Which layers of BERT are used for initialization? BERT has 12 layers but the text encoder only has 4.
* Without ground truth mention boundaries, Lee et al., 2017 (which is failing to detect mentions) isn’t a fair comparison - I’d suggest removing it from the table altogether as it is more distracting/confusing (feel free to leave a note in the appendix that it was tried but had low scores, possibly because there were no gold mentions).

**Reasons To Accept:**

* The paper proposes a new semi-supervised model for vision-and-text for the task for coreference grounding and shows that it outperforms some off-the-shelf several baselines or previously reported work.
* They thoroughly study the model, along with ablations and additional experiments/analysis to justify the complexity of the model.

**Reasons To Reject:**

* The task/dataset is still new/private (I can't find it online) and so there aren't really any established baselines on this dataset. Therefore, strong baselines are quite important. In particular, a pipelined approach of fine-tuning a neural coref model on the narrations (as opposed to using them out of the box) and then integrating into ALBEF (or even this paper's model) would be a fairer comparison + stronger starting point for CIN, since the proposed model also gets to fine-tune on that data.

**Reproducibility:**

2: Would be hard pressed to reproduce the results. The contribution depends on data that are simply not available outside the author's institution or consortium; not enough details are provided.

**Reviewer Confidence:**

3: Pretty sure, but there's a chance I missed something. Although I have a good feel for this area in general, I did not carefully check the paper's details, e.g., the math, experimental design, or novelty.

**Typos Grammar Style And Presentation Improvements:**

* L237: “where, softmax” -> “where the softmax”
* L376: This should be “Devlin et al., 2018” (or no citation since it is mentioned in the sentence already). It should definitely not be “Vaswani et al., 2017.”
* **Please make Table 2 and Table 3 bigger**. Also it is confusing why 50% labeled is not scoring the same as 50% unlabeled? What am I not understanding here/do they not add up to 1?
* **And all the tables really should be bigger/more readable.** You can mess with the hspace between columns if it means the font for the numbers can be bigger. Or consider moving full results (with MUC/B^3/CEAF) into the appendix and only report CoNLL F1 in the main body (except for Table 1: the subscores are quite interesting there!)
* L588: “The supervised contrastive loss effectives promotes…more distinctive embeddings for the clusters” – is this proven, or just a hypothesis that this is what is happening? This claim feels a bit strong here when the whole system is more complex than just CR loss, but I could be misunderstanding the phrasing in this section.
* Eq(1) is just cosine similarity right? It doesn't need an equation and you can save space.
* Similarly, all the softmax equations can be \softmax(...) instead of a big fraction (as long as it's clear what the softmax is over)

---

> ### Author Rebuttal · Authors · 2023-08-28
>
> We thank the reviewer for the valuable comments which we respond to below.
>
> > The task/dataset is still new/private (I can't find it online) and so there aren't really any established baselines on this dataset. Therefore, strong baselines are quite important.
>
> Prior to the submission deadline, we contacted the authors for a copy of the dataset which is now available at https://github.com/VICO-UoE/CIN along with the accepted ICCV’23 preprint. We agree that this is still a new task and dataset. Hence we included new uni-modal (longdoc) and multi-modal (VisualBERT, UNITER, ALBEF) baselines over the original work as discussed in Table 1.
>
> > In particular, a pipelined approach of fine-tuning a neural coref model on the narrations (as opposed to using them out of the box) since the proposed model also gets to fine-tune on that data.
>
> As suggested, we now provide a new neural coref baseline for a fair comparison that fine-tunes the Longformer-large model (Toshniwal et al. 2021) only on the labeled set of the CIN dataset. Its weights are initialized with the pre-trained model on other known 8 coref datasets including OntoNotes, PreCo. Surprisingly, it is nontrivial to finetune this large model without overfitting into the small labeled set of the CIN. The CoNLL F1 score for the fine-tuned model is 42.41 which is slightly lower than the score of 45.31 from the pre-trained model (Table 1, 2nd row). This also emphasizes the need for learning from larger unlabeled data. We will add these results to the paper and discuss them.
>
> > then integrating into ALBEF (or even this paper's model) would be a fairer comparison + stronger starting point for CIN.
>
> A shortcoming of finetuning coref models such as Longformer-large that are trained only on text data is their inability to provide associations with the visual input. Hence, finetuning Longformer-large yields lower performance (42.41) than even our self-supervised model with a F1 of 57.19 (Table 4, row 1) which is learned originally on multi-modal dataset.
>
> > Why doesn’t the grounding model use the ground-truth bounding boxes during training, or at least parts of training (L287-289) for g? This could be viewed as some kind of teacher forcing and should be an objective if it is available.
>
> Thanks for the valuable suggestion. Due to the short rebuttal time and required computation time, we leave this as future work where we would also like to take this a step further by jointly learning the bounding box prediction for mentions along with the coref.
>
> > L421: Which layers of BERT are used for initialization? BERT has 12 layers but the text encoder only has 4.
>
> After careful experimentation, we find that initializing with the first 4 layers of BERT works best. Thanks for raising this and we will clarify this in the final version.
>
> > Without ground truth mention boundaries, Lee et al., 2017 (which is failing to detect mentions) isn’t a fair comparison - I’d suggest removing it from the table altogether as it is more distracting/confusing (feel free to leave a note in the appendix that it was tried but had low scores, possibly because there were no gold mentions).
>
> Thanks, we will move this result to the appendix and replace it with a footnote in the final version.
>
> > L588: “The supervised contrastive loss effectives promotes…more distinctive embeddings for the clusters” – is this proven, or just a hypothesis that this is what is happening? This claim feels a bit strong here when the whole system is more complex than just CR loss, but I could be misunderstanding the phrasing in this section.
>
> Thanks for pointing this out. This is a hypothesis and we will replace L588 as follows: We hypothesize that the supervised contrastive loss provides a better clustering of mentions by contrasting them in the embedding space directly than the binary cross-entropy loss.
>
> > Eq(1) is just cosine similarity right? It doesn't need an equation and you can save space.
>
> Thanks, we will edit this.
>
>
> > Grammar and presentation improvements
>
> Thanks for the suggestions, we will update these.

---

### Meta-Review · Area_Chair_fE5K · 2023-09-18

**Recommendation:** 5

**Metareview:**

All reviewers felt that the work was both sound and exciting. In particular, there was general agreement that the proposed model, and thorough experiments, show a convincing improvement over strong results from previous work. Reviewers had a few points about baselines and clarity of motivation, which the author response addressed satisfactorily.

---

### Decision · Program_Chairs · 2023-10-07

**Decision:**

Accept-Main

**Comment:**

All reviewers felt that the work was both sound and exciting. In particular, there was general agreement that the proposed model, and thorough experiments, show a convincing improvement over strong results from previous work. Reviewers had a few points about baselines and clarity of motivation, which the author response addressed satisfactorily.